# Impact of health systems interventions in primary health settings on type 2 diabetes care and health outcomes among adults in West Africa: A systematic review protocol

Eugene Paa Kofi Bondzie[1]*, Kezia Amarteyfio[1], Yasmin Jahan[2ᵒ], Dina Balabanova[2ᵒ], Tony Danso-Appiah[3], Tolib Mirzoev[2‡], Edward Antwi[4], Irene Ayepong[1‡]

1 Ghana College of Physicians and Surgeons, Accra, Ghana, 2 London School of Hygiene and Tropical Medicine, London, United Kingdom, 3 University of Ghana, Legon, Accra, Ghana, 4 Ghana Health Service, Accra, Ghana

ᵒ These authors contributed equally to this work.
‡ TM and IA also contributed equally to this work.
* eugenebondzie@gmail.com

## Abstract

Type 2 diabetes is a major global public health challenge, particularly in the African region. Though evidence exists on pharmacological agents and non-pharmacological interventions in maintaining blood glucose concentration, the healthcare systems' ability to meet patients' needs may be inadequate. The management of non-communicable diseases, particularly diabetes, has been postulated to depend on functioning health systems. This systematic review will, therefore, summarize the current evidence on existing health systems interventions in primary health settings for type 2 diabetes care and health outcomes in West Africa and will explore the impact of these system-level interventions on service availability, accessibility and quality, as well as individualized outcomes such as glycemic control, disease awareness and treatment adherence. The review will be conducted according to the reporting guidance in the Preferred Reporting Items for Systematic Reviews and Meta-Analyses Protocols (PRISMA-P). The health system framework by Witter et al., 2019 will guide the system-level interventions and the search strategy to be explored in this review. This framework was designed to integrate the six building blocks of the World Health Organization (WHO) health systems framework and it delineates how they work synergistically to improve specific health outcomes. We will search the following databases PubMed, Google Scholar and Cumulated Index to Nursing and Allied Health Literature (CINAHL) between January 2000 to January 2024. We will also search Cairn.info for articles published between January 2005 to January 2024. The Cochrane Collaboration tool for assessing Risk of Bias will be implemented in each included study. We will conduct a narrative synthesis and make comparisons across findings using Excel-generated tables. **In Conclusion,** this systematic review will outline the existing system-level interventions that aim to or already improve type 2 diabetes services in primary health settings in West Africa and will offer suggestions for the strengthening and co-production of successful interventions that can be generalized to the entire sub-region.

**Data Availability Statement:** No datasets were generated or analysed during the current study. All relevant data from this study will be made available upon study completion.

**Funding:** This research was conducted as part of the health research on strengthening of capacity for Non-communicable disease (NCD control in West Africa (Stop NCDs project) commissioned by the National institute for health Research (NHIR), global health research centres: research and institutional capacity strengthening In NCDs Call 1 (grant number- 203246). NIHR Grange House 15 Church Street Twickenham TW1 3NL Tel: 020 8843 8000 Email: ccf@nihr.ac.uk www.nihr.ac.uk NO- The funders did not and will not have a role in study design, data collection and analysis, decision to publish, or preparation of the manuscript.

**Competing interests:** The authors have declared that no competing interests exist.

## Introduction

Type 2 diabetes, formerly known as adult-onset diabetes, is a major global public health problem [1]. It is defined as a form of diabetes mellitus characterized by high blood glucose, insulin resistance and a relative lack of insulin. It manifests with symptoms of increased thirst, frequent urination, weight loss and sometimes increased hunger [2]. Long-term complications from high blood sugar include ischemic heart disease, retinopathy, nephropathy and limb amputations [3]. Maintaining blood glucose concentration is highly essential to prevent severe complications [4]. Until recently, it was thought to affect adults who were middle-aged, or older but contemporary trends have shown an increase in incidence in young people [5]. Reports from epidemiological studies suggest that adults aged 20–79 years living with type 2 diabetes are about 10.5% of the world's population [6]. In Sub-Saharan Africa, the disease is estimated to increase by 129% by 2045 due to an increase in population and rapid urbanization [6]. This phenomenon is worsened by the considerable number of undiagnosed individuals living with the disease. According to published reports from the International Diabetes Federation (IDF) Atlas, people with undiagnosed diabetes in the African region represent the highest comparative proportions worldwide, currently at 54% and expected to increase by 2045 [6]. Consequently, the number of diabetes-related deaths was reported at 416,000 compared with 111,100 deaths in Europe. These disproportionate numbers reflect the poor structures at various levels of the primary care management of non-communicable diseases (NCDs) in the African region [7, 8]. West African countries have estimated prevalence rates of 3.7% in Nigeria [9], 3.95% in Ghana [10] and 1.7% in Burkina Faso [11]. Despite these numbers, glycemic control is suboptimal [12–14].

Evidence exists on the effectiveness of pharmacological and non-pharmacological treatment for type 2 diabetes [15–17] but the health systems' ability to meet patients' needs may be inadequate [2, 7].

The management and control of type 2 diabetes is largely dependent on health systems [18]. The treatment of diabetes is challenging because health systems are designed for short-term care rather than long-term care of people [19]. A healthcare system is an organization of people, institutions, and resources that deliver healthcare services to meet the health needs of target populations [20].

Effective health systems can improve the delivery of diabetes care and promote patients' access and use of quality services including access to medications, health facilities and specialists, with an overall impact on glycemic control and other associated outcomes (e.g., adherence, morbidity and mortality etc.) [20–22]. The primary healthcare system in West Africa is oriented towards the management of communicable diseases. Whereas NCD management appears to be less effective [23, 24]. Healthcare policies and intervention programs mainly exist on paper in many parts of the region [25–27]. Primary healthcare facilities are ill-prepared to implement essential interventions for type 2 diabetes control. Results from assessment surveys show a lack of essential medicines, basic equipment and diagnostics; lack of qualified trained personnel; and poor referral systems for the management of diabetes [28–31]. An earlier systematic review was conducted by Nuche-Berenguer, et al. [32] on the readiness of health systems to tackle diabetes in Sub-Saharan Africa, with part of the review focused on evaluating pilot projects, targeted at enhancing the capacity of health systems in diabetes care. However, this review was conducted six years ago, at a time when health system interventions towards type 2 diabetes were now evolving in most Sub-Saharan African countries, especially West African countries. Thus, most of the studies found in that review were based in eastern and southern Africa and the pooled findings may not be generalizable to other parts of Africa. The

review also could not synthesize the impact of the interventions on glycemic control due to the scarcity of relevant papers.

However, studies have been conducted between that period and the present to find the impact of health system-strengthening interventions on glycemic control. These include strategies aimed at improving the training and distribution of healthcare personnel, improving patient follow-up appointments, improving medicines distribution, and decentralizing services to patients at the primary-care level etc. Also, diversity exists between the subpopulations, their economic growth, health facility distribution, general disease burden and political structures. Thus, it will be important to find which interventions have been practical in the West African subpopulation.

According to the World Health Organization (WHO), health systems interventions can be defined as any array of initiatives that improves one or more of the functions of the health systems and that leads to better health through improvements in access, coverage, quality or efficiency [33]. Chee et al. [34] proposed that these initiatives should aim at permanently making the systems function better, not just filling gaps or supporting the systems to produce better short-term outcomes. They further submit that an intervention to strengthen the health system should go beyond providing inputs and apply to more than one building block, citing the six core components of health systems by the WHO. For this review, the health system interventions framework by Witter et al. [33] in 2019 would guide the system-level interventions to be explored (Fig 1). This framework was designed to integrate the six building blocks of the WHO health systems framework [20]. The framework by Witter et al. describes mechanisms of change within six health system blocks (governance, financing, infrastructure, workforce, supply chain and information), describes the implementation process goals and outlines the final desired outcomes. Interventions that promote any of these domains of the health system or across the integration of the domains have been proven to improve health outcomes in Africa [35].

This systematic review will summarize the recent evidence on health system interventions implemented in primary care settings to improve the accessibility, delivery and quality of healthcare for type 2 diabetes in West Africa; and will explore the impact and effectiveness of these interventions on overall glycemic control, disease awareness and treatment adherence.

## Review question

This systematic review aims to answer the question;

How do health system interventions influence healthcare in primary care settings and health outcomes of type 2 diabetes among adults in West Africa?

## Objectives

1. To identify the health system interventions in primary health settings that influence the accessibility, delivery and quality of type 2 diabetes care among adults in West Africa.

2. To synthesize the evidence on the effectiveness of these system-level interventions on glycemic control, awareness and treatment adherence.

3. To explore the impact of these system-level interventions on any associated health outcomes (i.e., any other patient outcomes or facility-level outcomes associated with an intervention).

## Methods

This systematic review will follow the reporting guidance provided in the Preferred Reporting Items for Systematic Reviews and Meta-Analyses Protocols (PRISMA-P) statement [36].

### Criteria for considering studies for this review

The PICOS (Participants, Interventions, Comparisons, Outcomes, Study types) eligibility criteria are shown in Fig 2.

### Types of participants and setting

- The participants to be covered in this review will be adults aged 18 years or more in West Africa, with a diagnosis of type 2 diabetes.

| Health system Inputs | Mechanisms of Change | Outcomes |
|---|---|---|
| Service delivery<br><br>Health Workforce<br><br>Supply chain<br><br>Health information systems<br><br>Financing<br><br>Governance | Applying a change or an intervention in each of the health system inputs either singly or in combination.<br><br>For example, training and skills building, changed incentives, social dialogue (physician-patient/ health provider-patient), improving clinic wait times, improving and maintain logistics, organizational culture change, new administrative procedures (e.g., governance and financing processes), structural reforms<br><br>Empowering people's power, interests and processes. | Quality, safe services available and accessible<br><br>Responsiveness<br><br>Efficiency (e.g., treatment outcome; good glycemic control)<br><br>High coverage of interventions<br><br>Reduced risk prevalence<br><br>Improvement in other health outcomes and equity<br><br>Social and financial risk protection |

**Fig 1. Health systems interventions framework (modified from the health systems strengthening framework by Witter et al. [33]).**

| Population | Adults aged 18 years or more in West Africa |
|---|---|
| Intervention | System-level interventions that targets the health system inputs in the Witter et al health system frame work within primary health facilities |
| Comparison | Corresponding groups where there were no strategies to enhance any of the health systems inputs with respect to diabetes care |
| Outcome | Main outcome is glycaemic control. Other health outcomes include: awareness, treatment adherence, type 2 diabetes services availability and patient access, affordability. |
| Study design | Randomized Controlled Trials (RCTs), Clinical Controlled trials (CCTs), quasi experiments |

**Fig 2. PICOS eligibility table.**

- Studies conducted in a primary health setting (community health and planning services (CHPS) compound, health center, district hospital/clinic) will be eligible.

- Studies on gestational diabetes (since its natural history varies significantly from type 2 diabetes [37]), and type 1 diabetes will be excluded.

- Also, studies that evaluated both type 1 diabetes and type 2 diabetes together will be excluded if we cannot extract findings from the group with type 2 diabetes.

### Types of interventions

We will define the health system interventions using the modification from the framework by Witter et al. [33] and a descriptive approach employed by Byiringiro et al. [35].

- Studies would be eligible if the interventions address the following health system inputs (i.e., service delivery, health workforce, supply chain, health information systems, financing governance) in the following ways;

- Service delivery. WHO defines service delivery as the activities that directly provide safe, effective, and high-quality health services to patients in need. Thus, we will consider the studies to enhance the capacity of this aspect of the health systems if the intervention was to enhance patients' access to health services for diabetes like screening, treatment, and follow-up, either through equitable distribution of care services, reduction of out-patient waiting time, revision of time allocated for services, or integrating the delivery of diabetes care with other established health services like HIV or TB.

- Health workforce. Interventions of interest in this aspect of the health systems would focus on healthcare providers who manage diabetes. The interventions include strategies to increase the number of providers, improve provider knowledge and implementation of diabetes management guidelines, address provider decision support systems, promote teamwork and institute task-sharing or task-shifting strategies to include providers who do not normally perform certain diabetes management tasks like the prescription of medications.

- Supply chain. In this aspect of the health systems, we will include studies with interventions that enhance the procurement systems to ensure the availability of anti-diabetic medications, availability and maintenance of calibrated glucometers, availability of consumables to conduct screening and other diabetes investigations, patient follow-up technologies like use of short message systems, and treatment guidelines.

- Health information systems. We will consider interventions to enhance the health information systems if they study an element of the process of patient data collection, analysis, sharing, and use to improve outcomes of interest. Interventions that explore the use of patient registries and information between patients and providers and among providers will be included.

- Financing. The financing aspect of the health systems, as defined in the WHO building blocks, has the most significant relevance on the national macro-level health systems. It also has an impact on the micro-level health systems' service delivery and patients' outcomes. We will consider studies where the interventions aimed at reducing patients' out-of-pocket spending or funding of diabetes care at health facilities through national and sub-national spending on health insurance premiums, or other relevant financial reliefs.

- Governance. This aspect has direct and indirect associations with the other five building blocks of health systems. We will consider studies to address leadership and governance if the intervention promotes a facility's leadership awareness of the burden of poor diabetes management, or if the intervention uses strategic planning and implementation of national diabetes management protocols in a healthcare facility, explores the accountability measures at the health facility, applies regular performance appraisal and planning for improvement, integrates supportive mentorship to lower primary health facilities, institutes a patient feedback collection and response system, or examines the leadership allocation of funds for diabetes management. We will also consider interventions where leadership joins or collaborates with other health facilities or local/international/national/private/non-governmental organizations to manage diabetes.

- Studies that piloted interventions with a minimum follow-up period of six months will be eligible.

- We will exclude studies that examined only patient-level or community-level lifestyle/behavioural changes or other non-conventional medical/non-medical interventions for diabetes management.

## Comparison

Intervention groups would be compared with corresponding groups where there were no strategies to enhance any of the health system areas for diabetes care.

## Outcome

We will evaluate the impact of the interventions on any of the following outcomes:

- Glycemic control. Defined by the American College of Endocrinologists as glycated haemoglobin (HbA1c %) levels below 7% or fasting blood sugar levels below 110mg/dl (6.1 mmol/l) [4].

- Diabetes awareness. Defined as persons with clinically measured diabetes (either with HbA1c or fasting blood sugar levels), who have been diagnosed by a physician or health provider with training to make a diagnosis.

- Treatment Adherence. Defined as consistently following a treatment plan (either an oral anti-diabetic or insulin or a lifestyle plan) as prescribed/advised/implemented by a healthcare provider [22].

- Any other associated health outcomes such as available services, financial risk protection, and reduced complications.

## Types of studies

- We will include studies with randomised controlled trials (RCTs) trials, clinical control trials (CCTs) and quasi-experimental designs.

- Observational studies including cross-sectional studies, cohort studies, case-control studies and case series will be excluded.

- Gray literature (e.g., books, commentary, dissertations, conference proceedings, modelling and simulation studies) will also be excluded.

## Language

We will include articles published in English or French. These two are the main languages spoken in West Africa.

## Search strategy

**Bibliographic databases.** We will search the following databases PubMed, Google Scholar and Cumulated Index to Nursing and Allied Health Literature (CINAHL) between January 2000 to January 2024. This period is selected to encompass studies that would have likely benefited from recent developments and improvements in RCTs, CCTs and quasi-experimental studies. We will also search Cairn.info for French articles published between January 2005 to January 2024.

We will define the search terms for primary healthcare of diabetes by borrowing and modifying the search strategy published in a systematic review of health systems interventions for hypertension by Byiringiro et al. [35] These search terms will likely encompass the primary healthcare practices for diabetes management in our setting of interest.

The search strategy consists of intersections between the following medical subject headings (MeSH); ("Health Services," Delivery of Health care," "Primary Health Care," "Health Facilities," "Health Care Facilities, Manpower, and Services," "Healthcare Financing," "Insurance, Health, Reimbursement, "Health Information Systems," "Equipment and Supplies"); and intersected with type 2 diabetes ("Diabetes mellitus, type 2," "Glycemic Control," "Diabetes Complications," "Hyperglycemia,") and West African countries ("African, Western"). The full search strategy is presented in (S1 File).

## Searching other resources

We will check the bibliographies of all relevant studies to find other potential publications. The references of any relevant systematic reviews and scoping reviews found will also be checked for relevant trials.

We will also contact the authors of articles whose full text is not easily accessible after searching through multiple online databases and where there are questions related to the results of the study or trial design, we will seek confirmation on the information that we extract from their studies. In case of no feedback from the authors, the corresponding studies will be excluded. Also, studies that do not measure any of the outcome variables (glycemic control, adherence, and awareness) will be excluded. Where there are duplicates for studies published in more than one paper, the most comprehensive one reporting the largest sample size will be considered.

## Study selection

Two reviewers will independently scan the titles and abstracts of all records yielded by the search to decide their eligibility for full-text screening and their full articles will be assessed through the databases and imported into the Rayyan screening software [38].

The full text of all potentially eligible articles will be screened and assessed for inclusion into the review by a trained reviewer, using a pre-specified eligibility form based on the inclusion criteria (S2 File). At least one paper will be piloted before the main selection. Disagreements will be resolved through discussion with a third reviewer. Neither of the review authors will be blind to the authors, institutions or journal titles of potential articles. We will present the number of included studies and the number of excluded studies using the PRISMA flow chart (Fig 3).

## Data extraction and management

Two reviewers will independently extract data from included studies and record them on pre-designed forms (S1 Table). The form would first be piloted on two included studies to ensure information is captured standardly. Disagreements will be resolved in consultation with the third reviewer.

## Dealing with missing data

We will contact the authors of included studies for which data related to study methods, and outcomes are unclear or missing.

## Risk of bias and quality assessment of included studies

To assess for possible risk of bias in the included studies, we will collect information using the Cochrane Collaboration tool for assessing the Risk of Bias [39] in a study (S3 File). The risk of bias tool would be implemented on each study and scored as high, low, or unclear risk (where there is insufficient detail reported), for the following domains: random sequence generation (selection bias); allocation concealment; blinding of participants and personnel (performance bias); blinding of outcome assessment (detection bias); incomplete outcome data (attrition bias); selective reporting (reporting bias). These scores would be assigned independently by the two review authors. Disagreements would be solved by discussion and consultation with a third reviewer. A table representing the risk of bias assessments within and across studies will be computed using Cochrane's Review Manager (Version 5.3).

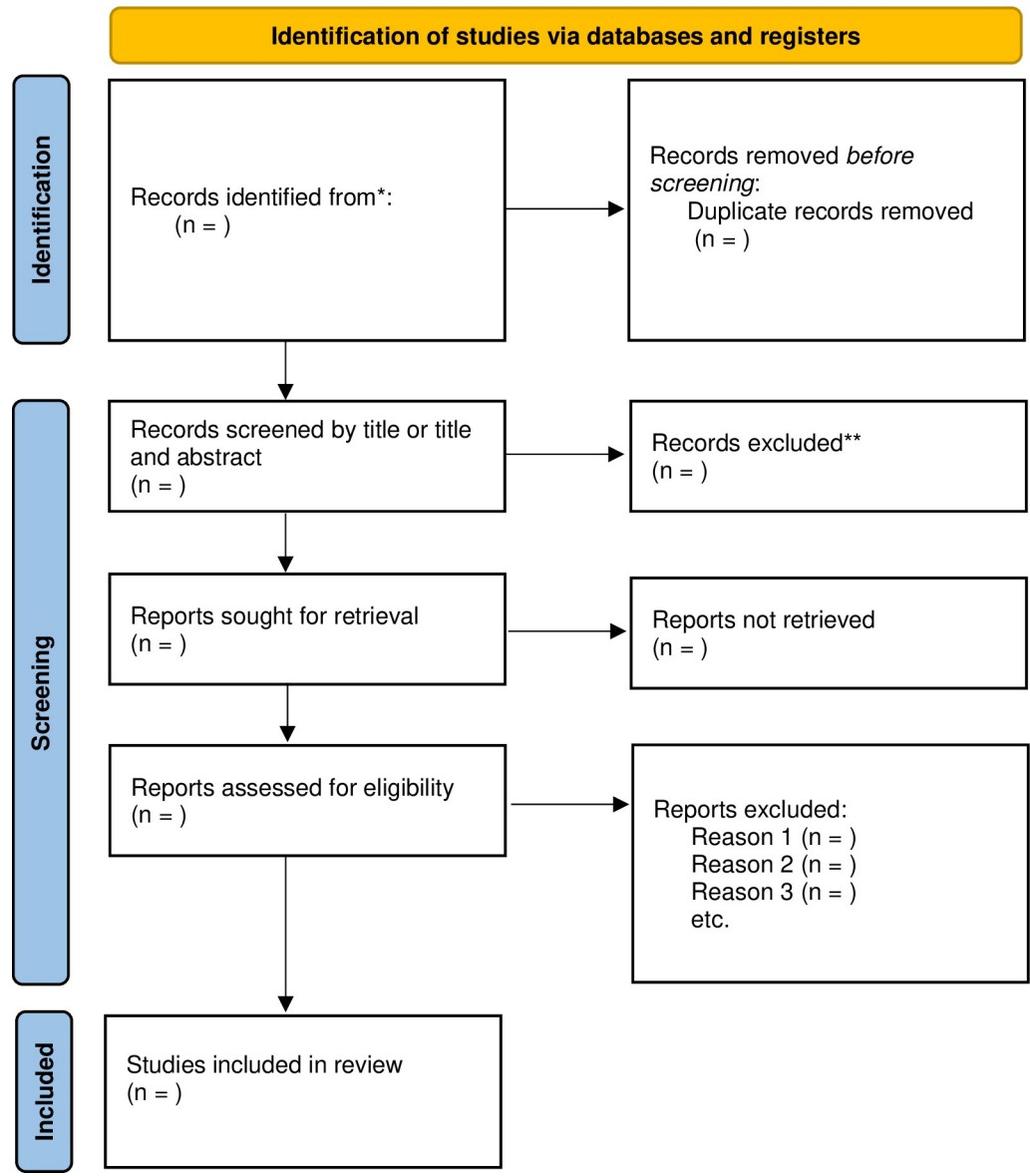

**Fig 3. PRISMA flow chart.**

## Data synthesis and analysis

We will conduct a narrative synthesis of the health system interventions found in the studies by creating a table to compare the PICO elements. We will classify the outcomes into the following groups; glycemic control, diabetes awareness and treatment adherence. We will use relative risk to report the effect measures of the interventions on diabetes outcomes and compare them across all included studies. Our findings will be reported using tables generated from Excel.

## Assessment of confidence in the cumulative estimate

The Grading of Recommendations, Assessment, Development, and Evaluation (GRADE) [40] approach will be implemented to assess the quality of the evidence for any of the outcomes.

We will report a table on the GRADE evidence profile for the quality of the studies on the following domains: 1) risk of bias 2) inconsistency 3) indirectness 4) imprecision 5) publication bias 6) magnitude of effect 7) residual 8) dose-response 9) overall GRADE quality scored as very low (very uncertain about the estimate of effect), low (further research is very likely to change the estimate of effect, moderate (further research may likely change the estimate of effect), high quality (i.e., further research is very unlikely to change our confidence in the estimate of effect).

## Ethical considerations

Ethical approval will not be needed for this review protocol because data will be extracted from published studies and there will be no concerns about privacy.

## Discussion

By the end of this review, we aim to provide a report on integrated system-level interventions in the health systems of primary health facilities aimed at improving the delivery and quality of type 2 diabetes services as well as patients' access to these services. We will further explore the effectiveness of these interventions on the impact on patients' glycemic control. We would then outline areas of strengths and gaps in already existing interventions for health policies, co-production of interventions and widespread uptake in the entire region.

### Strengths and limitations

This review will offer valuable insights into the latest health system interventions for the management of type 2 diabetes in West Africa and explore the effect of integrated system-level interventions on the provision of care for type 2 diabetes. The literature search would be conducted in only English and French thus we are likely to miss relevant studies in other languages. E.g., Portuguese.

### Dissemination plans

Results from this review would be disseminated through academic journals, policy briefs, and stakeholder and intervention co-production workshops. All results will be made fully available upon completion of this study.

### Dealing with amendments

The corresponding author, Eugene Paa Kofi Bondzie, is responsible for any amendments and updates of this review protocol.

## Supporting information

**S1 File. Search strategy.**
(PDF)

**S2 File. Study selection flow chart.**
(PDF)

**S3 File. Cochrane risk of bias tool.**
(PDF)

**S1 Table. Data extraction form.**
(DOCX)

## Acknowledgments

We would like to acknowledge Mary Pomaa Agyekum, the librarian who helped in the development of the search strategy.

## Author Contributions

**Conceptualization:** Eugene Paa Kofi Bondzie, Yasmin Jahan, Dina Balabanova, Irene Ayepong.

**Funding acquisition:** Tolib Mirzoev, Irene Ayepong.

**Methodology:** Eugene Paa Kofi Bondzie, Yasmin Jahan, Dina Balabanova, Tolib Mirzoev, Irene Ayepong.

**Supervision:** Dina Balabanova, Tony Danso-Appiah, Tolib Mirzoev, Edward Antwi, Irene Ayepong.

**Validation:** Dina Balabanova, Tolib Mirzoev, Irene Ayepong.

**Visualization:** Eugene Paa Kofi Bondzie.

**Writing – original draft:** Eugene Paa Kofi Bondzie.

**Writing – review & editing:** Eugene Paa Kofi Bondzie, Kezia Amarteyfio, Yasmin Jahan.

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
