## [Decision Letter · Decision Letter 0]

1 Feb 2024

PONE-D-23-26974Impact of health systems interventions in primary health settings on type 2 diabetes care and health outcomes among adults in West Africa: a systematic review protocolPLOS ONE

Dear Dr. Bondzie,

Thank you for submitting your manuscript to PLOS ONE. After careful consideration, we feel that it has merit but does not fully meet PLOS ONE’s publication criteria as it currently stands. Therefore, we invite you to submit a revised version of the manuscript that addresses the points raised during the review process.

We look forward to receiving your revised manuscript.

Kind regards,

Hubert Amu

Academic Editor

PLOS ONE

Journal Requirements:

Additional Editor Comments:

I kindly invite the authors t address the minor comments made by the reviewers to improve the quality of this manuscript before it is accepted for publication in Plos One.

Reviewers' comments:

Reviewer's Responses to Questions

**Comments to the Author**

1. Does the manuscript provide a valid rationale for the proposed study, with clearly identified and justified research questions?

Reviewer #1: Partly

Reviewer #2: Yes

2. Is the protocol technically sound and planned in a manner that will lead to a meaningful outcome and allow testing the stated hypotheses?

Reviewer #1: Yes

Reviewer #2: Yes

3. Is the methodology feasible and described in sufficient detail to allow the work to be replicable?

Reviewer #1: Yes

Reviewer #2: Yes

4. Have the authors described where all data underlying the findings will be made available when the study is complete?

Reviewer #1: No

Reviewer #2: Yes

5. Is the manuscript presented in an intelligible fashion and written in standard English?

Reviewer #1: Yes

Reviewer #2: Yes

6. Review Comments to the Author

You may also provide optional suggestions and comments to authors that they might find helpful in planning their study.

Reviewer #1: Authors Bonzie et al, have written a nicely crafted protocol for a SR on an important topic. Congratulations. Below are some observations which I feel would help the protocol to improve in what it intends to serve.

Title: I would rather choose effectiveness over impact in the title.

Financial Disclosure: Researchers mention that this research was conducted …… While this is just a protocol for a SR, is that correct to mention to write that way. Is this SR funded with that grant, is this SR a part of another big study or some project. Please revisit and mention it clearly what exactly it is and mention it in the right way.

Background:

Please check referencing. Ref no 6 for e.g., has been provided with an inappropriate way in the references, please check the authors you have written for this reference and correct it.

Adding the burden of disease information of Type 2 DM would strengthen the background, rather than just the proportion of people with DM in the region. Also, you could narrow down the information to West Africa where you plan to focus your SR on.

Till line 107, the authors talk about sub–Saharan Africa without any mention of west Africa, and then in the next paragraph line 108 – 114, the authors suddenly mention the need of the study in West Africa. We do not find a clear transition to this para. In addition, the authors have probably failed to illustrate the current health system situation. A clear and coherent information about the disease burden and health care delivery system situation including availability and readiness with a focus on west African region would set a clear background information for the proposed SR.

Research Questions:

I would rephrase the research question as below:

How do health systems interventions influence healthcare in primary care settings and health outcomes of type 2 diabetes among adults in West Africa?

Objectives:

The way the objectives are written, it looks like general and specific objectives appear to be written as if they are objective 1, 2 & 3. Please have a look once again. Provide a more general/overall objective in general objective, if you would like to follow this structure of general and specific objectives. As per your research question and topic of SR, current general objective cannot be your general objective. It just says to identify the interventions…… is that what your SR is all about?

Please reorganize the objectives section. Furthermore, in the last objective you mention associated health outcomes but while trying to qualify it as example you mention service availability, service utilization….. do you call these as outcomes??

Minor comments on methods:

While describing the PICOS, I would follow the order as in the table.

Thank you and all the best!!

Reviewer #2: suggest to add one more review question: Does an improvement on health outcomes could be observed among the population with various socioeconomic disparities ?

7. PLOS authors have the option to publish the peer review history of their article (what does this mean?). If published, this will include your full peer review and any attached files.

Reviewer #1: **Yes: **Krishna Kumar Aryal

Reviewer #2: No

---

## [Author Response · Author response to Decision Letter 0]

13 Feb 2024

REVIEWERS’ COMMENTS AND RESPONSES

Reviewer #1:

“Authors Bondzie et al, have written a nicely crafted protocol for a SR on an important topic. Congratulations. Below are some observations which I feel would help the protocol to improve in what it intends to serve.

Title: I would rather choose effectiveness over impact in the title”.

Response:

Thank you for the suggestion. We opted for impact over effectiveness in the title because whereas effectiveness often refers to the consequence of an action and concerned with the most closely attributable results, impact on the other hand implies the influence of an action on broader changes and effects. E.g. Effectiveness of Metformin on blood glucose explores the direct consequence of metformin on blood glucose. Our study seeks to identify the influence of health system interventions on both intrinsic, and final healthcare outcomes for type 2 diabetes. Also, acknowledging that biomedical interventions and life style interventions have influence on disease outcomes.

“Financial Disclosure: Researchers mention that this research was conducted …… While this is just a protocol for a SR, is that correct to mention to write that way. Is this SR funded with that grant, is this SR a part of another big study or some project. Please revisit and mention it clearly what exactly it is and mention it in the right way.”

Response:

Thank you for your comment. We have removed the financial disclosure from the manuscript per the advice of the editor. However, this systematic review will be conducted as part of the National Institute of Health and Research’s West African project on Non-Communicable diseases. The financial disclosure has been corrected on the online submission form.

“Background:

Please check referencing. Ref no 6 for e.g., has been provided with an inappropriate way in the references, please check the authors you have written for this reference and correct it.”

Response:

Thank you for pointing this out. Reference no. 6 has been corrected on line 391-392

IDF_Atlas_10th_Edition_2021.pdf [Internet]. [cited 2023 Jun 19]. Available from: https://diabetesatlas.org/idfawp/resource-files/2021/07/IDF_Atlas_10th_Edition_2021.pdf”-

“Adding the burden of disease information of Type 2 DM would strengthen the background, rather than just the proportion of people with DM in the region. Also, you could narrow down the information to West Africa where you plan to focus your SR on.

Till line 107, the authors talk about sub–Saharan Africa without any mention of west Africa, and then in the next paragraph line 108 – 114, the authors suddenly mention the need of the study in West Africa. We do not find a clear transition to this para. In addition, the authors have probably failed to illustrate the current health system situation. A clear and coherent information about the disease burden and health care delivery system situation including availability and readiness with a focus on west African region would set a clear background information for the proposed SR.”

Response:

Thank you for your suggestion. We have added the following lines to highlight the burden of type 2 diabetes in West Africa:

Line 78-80

“West African countries have estimated prevalence rates of 3.7% in Nigeria, 3.95% in Ghana and 1.7% in Burkina Faso. Despite these numbers, glycemic control remains suboptimal”

We also added the following lines to highlight the health care delivery system situation for Non-communicable diseases in West Africa:

Line 90-96

“The primary healthcare system in West Africa is rather oriented towards the management of communicable diseases. Whereas, NCD management appears to be less effective. Healthcare policies and intervention programs mainly exist on paper in many parts of the region. Primary health care facilities are ill prepared to implement essential interventions for type 2 diabetes control. Results from assessment surveys indicate lack of essential medicines, basic equipment and diagnostics; lack of qualified trained personnel; and poor referral systems for the management of diabetes.”

“Research Questions:

I would rephrase the research question as below:

How do health systems interventions influence healthcare in primary care settings and health outcomes of type 2 diabetes among adults in West Africa?”

Response:

Thank you very much for your suggestion. We have rephrased our research question to:

“How do health systems interventions influence healthcare in primary care settings and health outcomes of type 2 diabetes among adults in West Africa?” - Line 139-140

“Objectives:

The way the objectives are written, it looks like general and specific objectives appear to be written as if they are objective 1, 2 & 3. Please have a look once again. Provide a more general/overall objective in general objective, if you would like to follow this structure of general and specific objectives. As per your research question and topic of SR, current general objective cannot be your general objective. It just says to identify the interventions…… is that what your SR is all about?

Please reorganize the objectives section. Furthermore, in the last objective you mention associated health outcomes but while trying to qualify it as example you mention service availability, service utilization….. do you call these as outcomes??”

Response:

Thank you for this thoughtful comment. We have re-organized the objectives. They have been written as objective 1, 2 & 3. We have also re-worded the associated outcomes in the last objective

Line 142-147

1. To identify the health system interventions in primary health settings that influence the, accessibility, delivery and quality of type 2 diabetes care among adults in West Africa.

2. To explore the evidence on the effectiveness of these system-level interventions on glycemic control, awareness and treatment adherence.

3. To explore the impact of these system-level interventions on any associated health outcomes (i.e. any other patient outcomes or facility level outcomes associated with an intervention).

“Minor comments on methods:

While describing the PICOS, I would follow the order as in the table.”

Response: 

Thank you for your suggestion.

We have rearranged the description of the PICOS to align with the order in Fig 2.

Reviewer #2: 

“suggest to add one more review question: Does an improvement on health outcomes could be observed among the population with various socioeconomic disparities?”

Response:

Thank you for your thoughtful suggestion. We have considered this particular objective. However, while we recognize the potential value of an additional question, we would like to maintain the original focus of the review as per the registered protocol.

EDITOR COMMENTS AND RESPONSES

1. Please ensure that your manuscript meets PLOS ONE’S style Requirements, including those for file naming.

Response:

Thank you. We followed the PLOS ONE style templates to write the manuscript and organized file naming. We uploaded all figures separately.

Response:

The funding statement has been removed from the manuscript.

Response:

We have made changes to reference list and mentioned these changes in the rebuttal letter.

4. Have the authors described where all data underlying the findings will be made available when the study is complete?

Reviewer #1: No

Reviewer #2: Yes

Response:

We have added the following lines to clarify this question: Line 354-356

All results will be made fully available upon completion of this study.

---

## [Decision Letter · Decision Letter 1]

7 Jun 2024

PONE-D-23-26974R1Impact of health systems interventions in primary health settings on type 2 diabetes care and health outcomes among adults in West Africa: a systematic review protocolPLOS ONE

Dear Dr. Bondzie,

Thank you for submitting your manuscript to PLOS ONE. After careful consideration, we feel that it has merit but does not fully meet PLOS ONE’s publication criteria as it currently stands. Therefore, we invite you to submit a revised version of the manuscript that addresses the points raised during the review process.

**Comments from reviewers can be found attached. I strongly advise the use of a professional editing service to correct all grammatical errors.**

We look forward to receiving your revised manuscript.

Kind regards,

Blessing Onyinye Ukoha-kalu, B.Pharm, M.Pharm, Ph.D

Guest Editor

PLOS ONE

Journal Requirements:

Reviewers' comments:

Reviewer's Responses to Questions

**Comments to the Author**

1. Does the manuscript provide a valid rationale for the proposed study, with clearly identified and justified research questions?

Reviewer #3: Yes

2. Is the protocol technically sound and planned in a manner that will lead to a meaningful outcome and allow testing the stated hypotheses?

Reviewer #3: Yes

3. Is the methodology feasible and described in sufficient detail to allow the work to be replicable?

Reviewer #3: Yes

4. Have the authors described where all data underlying the findings will be made available when the study is complete?

Reviewer #3: Yes

5. Is the manuscript presented in an intelligible fashion and written in standard English?

Reviewer #3: No

6. Review Comments to the Author

You may also provide optional suggestions and comments to authors that they might find helpful in planning their study.

**Reviewer #3: **The use of 'However' in line 34 should be reconsidered in relation to the transition from the previous sentence.

Line 53 and 54, I don't think this should be included in abstract but in main text.

Line 49: This part of the sentence is not clear 'and Car.info from inception to January 2024'.

Check typos throughout, for example line 77/78 'These disproportionate number reflect the poor structures' should be numbers.

Most of the information is there but there is repetitiveness and the paper could benefit from a language sweep overall to strengthen the presentation.

7. PLOS authors have the option to publish the peer review history of their article (what does this mean?). If published, this will include your full peer review and any attached files.

Reviewer #3: No

---

## [Author Response · Author response to Decision Letter 1]

9 Jun 2024

REVIEWERS’ COMMENTS AND RESPONSES 

Reviewer #3: 

1. “The use of ‘However’ in line 34 should be reconsidered in relation to the transition from the previous sentence.” 

Response: 

Thank you for this comment. We have removed the word “However” from line 34. 

2. “Line 53 and 54, I don’t think this should be included in abstract but in main text.” 

Response: 

Thank you for your suggestion. We have removed lines 53 and 54 from the abstract. The limitations of the study have been maintained in the main text of the manuscript. 

3. “Line 49: This part of the sentence is not clear ‘and Car.info from inception to January 2024’ 

Response 

Thank you for your comment. The sentence has been re-worded as “We will also search Cairn.info for articles published between January 2005 to January 2024.” 

4. “Check for typos throughout, for example line 77/78 ‘These disproportionate number reflect the poor structures’ should be numbers. 

Most of the information is there but there is repetitiveness and the paper could benefit from a language sweep overall to strengthen the presentation. 

Response :

Thank you for your thoughtful suggestion. We have corrected all identified typos within the manuscript and removed repeated texts.

---

## [Editor Report · Decision Letter 2]

11 Jun 2024

Impact of health systems interventions in primary health settings on type 2 diabetes care and health outcomes among adults in West Africa: a systematic review protocol

PONE-D-23-26974R2

Dear Dr. Bondzie,

We’re pleased to inform you that your manuscript has been judged scientifically suitable for publication and will be formally accepted for publication once it meets all outstanding technical requirements.

Kind regards,

Blessing Onyinye Ukoha-kalu, B.Pharm, M.Pharm, Ph.D

Guest Editor

PLOS ONE
---

## [Editor Report · Acceptance letter]

9 Aug 2024

PONE-D-23-26974R2 

PLOS ONE

Dear Dr. Bondzie, 

I'm pleased to inform you that your manuscript has been deemed suitable for publication in PLOS ONE. Congratulations! Your manuscript is now being handed over to our production team.

Kind regards, 

on behalf of

Dr Blessing Onyinye Ukoha-kalu 

Guest Editor

PLOS ONE